# Mobile Spatiotemporal Gait Segmentation Using an Ear-Worn Motion Sensor and Deep Learning

**DOI:** 10.3390/s24196442

**Published:** 2024-10-04

**Authors:** Julian Decker, Lukas Boborzi, Roman Schniepp, Klaus Jahn, Max Wuehr

**Affiliations:** 1German Center for Vertigo and Balance Disorders (DSGZ), LMU University Hospital, 81377 Munich, Germany; 2Schön Klinik Bad Aibling, 83043 Bad Aibling, Germany; 3Institut für Notfallmedizin und Medinzinmanagement (INM), LMU University Hospital, 80336 Munich, Germany; 4Department of Neurology, LMU University Hospital, 81377 Munich, Germany

**Keywords:** gait analysis, inertial sensor, ear, earables, in-ear sensing, vital-sign monitoring, wearables, deep learning

## Abstract

Mobile health technologies enable continuous, quantitative assessment of mobility and gait in real-world environments, facilitating early diagnoses of gait disorders, disease progression monitoring, and prediction of adverse events like falls. Traditionally, mobile gait assessment predominantly relied on body-fixed sensors positioned at the feet or lower trunk. Here, we investigate the potential of an algorithm utilizing an ear-worn motion sensor for spatiotemporal segmentation of gait patterns. We collected 3D acceleration profiles from the ear-worn sensor during varied walking speeds in 53 healthy adults. Temporal convolutional networks were trained to detect stepping sequences and predict spatial relations between steps. The resulting algorithm, mEar, accurately detects initial and final ground contacts (F1 score of 99% and 91%, respectively). It enables the determination of temporal and spatial gait cycle characteristics (among others, stride time and stride length) with good to excellent validity at a precision sufficient to monitor clinically relevant changes in walking speed, stride-to-stride variability, and side asymmetry. This study highlights the ear as a viable site for monitoring gait and proposes its potential integration with in-ear vital-sign monitoring. Such integration offers a practical approach to comprehensive health monitoring and telemedical applications, by integrating multiple sensors in a single anatomical location.

## 1. Introduction

Gait evaluation is a well-established clinical tool used to assess general health, detect and differentiate diseases, monitor disease progression, and predict adverse events such as recurrent falls [1,2,3]. Traditionally, gait evaluation is performed in supervised, standardized clinical settings using qualitative or semi-structured assessments (clinical scores) or, in the best cases, apparatus-based measurements [4,5,6]. However, these assessments can only provide a snapshot of a patient’s mobility, potentially missing episodic or rare events like freezing of gait or falls [7]. Additionally, the ecological validity of these assessments is often questioned, as walking in a clinical setting can differ significantly from walking in daily life [8,9]. Consequently, there is growing interest in mobile health technologies that enable continuous, quantitative assessment of mobility and gait in real-world environments [10]. Previous research indicates that these continuous, unsupervised assessments may offer complementary, or even more accurate, insights for predicting disease progression or the risk of future falls [11,12].

Health technologies for continuous long-term monitoring of gait and mobility primarily rely on mobile sensors, which typically integrate accelerometers, often augmented with gyroscopes and magnetometers. These sensors’ small size and low energy consumption allow them to be attached to various body locations or to be integrated into wearable devices, such as smartwatches. Selecting the number and placement of these sensors is crucial to balancing the accuracy of gait-related motion detection with the need for unobtrusiveness to ensure long-term user engagement and adherence. Previous research suggests that attaching two or three motion sensors to the legs and lower trunk provides the most accurate estimation of spatiotemporal stepping events [13,14]. In contrast, motion sensors on the wrist, where wearables are commonly worn, can only provide imprecise information about gait-related data, often limited to basic step counting [15].

Recently, the head, particularly the ear, has emerged as a promising alternative site for mounting a single motion sensor to monitor mobility and gait [16,17]. Unlike the lower extremities or trunk, the ear offers several unique advantages for implementing an accurate and unobtrusive mobility monitor. First, the head—which houses crucial sensory systems for vision, hearing, and balance—remains exceptionally stable during various movements [18,19]. This stability provides a reliable location for low-noise identification and differentiation of different bodily activities. Research has demonstrated that a single ear-worn motion sensor can reliably distinguish between a wide range of daily activities and provide detailed insights into the number and temporal sequence of stepping events [16]. Additionally, the ear is a location where users, especially the elderly, often already use assistive devices like hearing aids or eyeglass frames, which can be easily combined with a miniature motion sensor. This integration could significantly reduce barriers to user-friendly long-term monitoring. Finally, the ear is an promising site for comprehensive vital-status monitoring using optical sensors that can measure pulse rate, blood pressure, oxygen saturation, and body temperature [20]. Combined with gait monitoring, often referred to as the sixth vital sign, a single ear sensor would offer a comprehensive overview of a patient’s physical health and motor condition.

In this study, we explore the potential of a mobile ear-based gait identification algorithm (mEar) to comprehensively identify spatiotemporal gait characteristics. For this purpose, we collected 3D acceleration profiles from the ear-worn sensor in a large cohort of healthy individuals walking at various paces, from slow to fast. These motion profiles were synchronized with a pressure-sensitive gait mat, which served as the ground truth for the temporal stepping sequence and the spatial characteristics of the gait pattern. We employed state-of-the-art deep-learning architectures to identify the spatiotemporal step sequence from the sensor data and evaluated the accuracy of these identifications at different levels of granularity, from walking bouts to individual ground contacts. Finally, we highlight the potential of the motion sensor integrated into a commercial, wearable in-ear vital-sign monitor for comprehensive and continuous monitoring of mobility, gait, and vital signs in patients’ daily lives.

## 2. Materials and Methods

### 2.1. Participants

Fifty-three healthy individuals, aged between 20 and 47 years (mean age: 29.9 ± 8.4 years, range: 20–59 years; height: 1.73 ± 0.10 m; weight: 70.1 ± 15.9 kg; 28 females), participated in this study. All participants provided written informed consent prior to inclusion and were screened for any neurological or orthopedic conditions that could affect balance or locomotion.

### 2.2. Ear-Worn Motion Sensor

The motion sensor (Figure 1A) consisted of a triaxial accelerometer (range: ±16 g; accuracy: 0.0002 g; sampling rate: 100 Hz), integrated into a commercial, wearable in-ear vital-sign monitor (c-med° alpha; dimensions: 55.2 mm × 58.6 mm × 10.0 mm; weight: 7 g; Cosinuss GmbH, Munich, Germany). The vital-sign monitor includes a silicone earplug that contacts the outer ear canal skin and contains an infrared thermometer for recording body temperature and an optical sensor for measuring pulse rate and blood oxygen saturation. The earplug is connected to an earpiece hooked around the ear conch, where the motion sensor is located (Figure 1A). The wearable device transmits acquired motion and vital signals in real time via Bluetooth Low Energy to a gateway, which then streams this information to the cosinuss° Health server.

### 2.3. Experimental Procedures

The participants walked across a 6.7 m pressure-sensitive gait mat (GAITRite^®^, CIR System, Sparta, NJ, USA), synchronized with the ear-worn motion sensor, which detected the spatiotemporal stepping sequence at 120 Hz (Figure 1B). They were instructed to walk at three different paces: preferred speed (“please walk at your normal walking pace”), slow speed (“please walk as slowly as possible while maintaining a fluid pace”), and fast walking speed (“please walk as quickly as you can without transitioning into jogging or running”). Each speed condition was repeated multiple times (slow: 6 times; preferred: 8 times; fast: 10 times) to gather sufficient gait cycles for estimating variability and asymmetry characteristics of the gait pattern [21].

The gait mat provided spatiotemporal gait characteristics used as ground truth for sensor training. Temporal stepping sequence was quantified by the timing of initial contacts (ICs) and final contacts (FCs) of the left and right feet, while spatial stepping sequence was defined by the x- and y-coordinates of the left and right feet at successive ICs. From these metrics, various gait cycle measures were obtained, including stride time (s), swing time (s), double support time (s), stride length (cm), and stride width (cm) (Figure 1C). These measures were evaluated at the walking-bout level in terms of quantifying the mean, variability, and asymmetry across all strides collected at a particular speed. Variability was assessed using the coefficient of variation (CV; 100× std/mean, %), and asymmetry was measured as 100× (1-mean(smaller foot value)/mean(larger foot value)) (%).

### 2.4. Gait Identification Model

#### 2.4.1. Model Architecture

A modified temporal convolutional network (TCN) architecture, previously demonstrated to be effective for sensor-based gait segmentation, was selected as the generic model scheme (Figure 2 and Appendix A) [13,14,22].

A TCN is composed of a sequence of residual blocks, with dilation factors that can increase exponentially. Each residual block typically includes layers such as dilated convolutional, batch normalization, ReLU activation, and dropout layers. The implemented TCN block for temporal gait event detection (tTCN; Appendix A) contains a single 1D convolutional layer, followed by batch normalization, a ReLU activation function, and a dropout layer. This convolutional layer uses a fixed kernel size, a stride of 1, and an adjustable dilation factor. Padding is applied to ensure the output sequence length matches the input, thereby preserving the temporal dimensions throughout the time series. In contrast, the implemented TCN block for determination of spatial gait characteristics (sTCN; Appendix A) consists of two consecutive 1D convolutional layers, each followed by ReLU activation and dropout. This block also includes padding removal (chomping) to maintain temporal causality and features a residual connection that allows the input to bypass the convolutional layers. This residual connection aids gradient flow, improving the training of deeper networks.

#### 2.4.2. Model Training

We trained a total of three neural networks: one for the detection of temporal gait events (temporal mEar) and two for the determination of spatial gait characteristics, namely, stride length and width (spatial mEar). The temporal mEar received raw triaxial accelerometer data (input size 3 × 200 samples) to identify the temporal gait events’ ICs and FCs. From the identified IC events, accelerometer values spanning a complete gait cycle were extracted and used as input for the regression of spatial stride characteristics with the two spatial mEar networks. All inputs were standardized using a robust scaler to remove the median and scale the data according to the interquartile range. For training and validation of the models, the datasets were initially split into training (80%) and test sets (20%) using a group-stratified split to prevent data from the same individual from appearing in multiple sets, thus avoiding potential bias or information leakage. The training set was further split into training and validation sets using a group-stratified k-fold cross-validation (with k = 5). Hypertuning of the model parameters (batch size, learning rate, and number of epochs) was performed using a common grid search strategy (for options and results, see Appendix A). Model training was performed with binary cross entropy (BCE) with logits or mean squared error (MSE) as a loss function, for the temporal or spatial mEar models, respectively. The Adam optimizer was used to iteratively learn the model weights. All models were built using Python 3.9 and PyTorch 2.3.1.

### 2.5. Performance Evaluation

The model performance was evaluated with respect to (1) detection performance and time agreement of predicted temporal gait events (i.e., ICs and FCs) with the ground truth (gait mat) and (2) the agreement of derived temporal and spatial stride parameters with the ground truth.

The overall detection performance measured the number of annotated events detected by the model (true positives, *TPs*), the number of annotated events missed by the model (false negatives, *FNs*), and the number of detected events that were not annotated (false positives, *FPs*). Using these metrics, detection performance war primarily evaluated by the weighted *F1 score*, which accounts for both precision and recall while adjusting for class distribution imbalances. It calculates the harmonic mean of *precision* and *recall* and ranges between 1 and 0, reflecting the best and worst performance, respectively:$$recall=\frac{TP}{TP+FN}$$
$$precision=\frac{TP}{TP+FP}$$
$$F1\;score=2*\;\frac{precision*recall}{precision+recall}$$

A detected event (either IC or FC) was considered a *TP* if the absolute time difference from the corresponding annotated event was <250 ms [13]. For all *TPs*, the time agreement with the ground truth was quantified via the following: $time\;error=t_{annotated}-t_{predicted}$.

We employed multiple statistical techniques to assess the agreement of derived temporal and spatial stride parameters with the ground truth, including the absolute and relative root-mean-square error (RMSE), Pearson’s correlation coefficient, and the intraclass correlation coefficient for absolute agreement (ICC(3,1); two-way mixed model). ICC outcomes were interpreted according to established categories [23]: poor agreement (<0.5), moderate agreement (0.5–0.75), good agreement (0.75–0.9), and excellent agreement (>0.9). All the above-mentioned metrics were calculated for the various gait parameters across all subjects and gait speeds. To examine potential differences in agreement at different speeds, the relative RMSE for each subject and speed was also calculated for all mean spatiotemporal gait parameters. Differences in RMSE results between speeds were tested using a repeated-measures analysis of variance (ANOVA). All statistical analyses were conducted using Python 3.9.

## 3. Results

### 3.1. Dataset Characteristics

A total of 2.59 h (left-worn sensors: 1.26 h; right-worn sensors: 1.33 h) of walking activity was recorded from the 53 participants. The collected dataset included a total of 2434 walks on the gait mat (563 at a slow walking speed; 864 at a preferred walking speed; 1007 at a fast walking speed), with a total of 11895 recorded gait cycles (3851 at a slow walking speed; 4193 at a preferred walking speed; 3851 at a fast walking speed).

### 3.2. Step Detection Performance

Table 1 displays the overall performance of the trained temporal mEar network to detect ICs (training samples: 14,600; test samples: 3650) and FC events (training samples: 12,428; test samples: 3107). Both types of events were identified with high accuracy, with the detection rate for ICs being nearly perfect and slightly less so for FCs (F1 score 99% vs. 91%). The time agreement between predicted and ground truth events was close zero for IC (3 ms) and for FC events (2 ms).

### 3.3. Accuracy of Temporal Gait Cycle Parameters

Based on the detected temporal gait events, various temporal gait cycle parameters (i.e., stride time, swing time, and double support time) were calculated. For each temporal gait cycle aspect, the mean, variability (i.e., CV), and side asymmetry were computed. Table 2 provides an overview of the agreement of these mEar-derived gait metrics with the gold standard. All mean temporal gait cycle parameters exhibited excellent agreement with the gold standard. Agreement for variability parameters was excellent in the case of stride time and dropped to good or moderate agreement for swing and double support time, respectively. Agreement for asymmetry parameters was only moderate in the case of stride time but poor for the remaining parameters.

### 3.4. Accuracy of Spatial Gait Cycle Parameters

Based on the temporally segmented gait cycles, two spatial mEar networks were trained to additionally estimate the spatial characteristics of walking (i.e., stride length and stride width; training samples: 10,256; test samples: 2564). For each spatial gait cycle aspect, the mean, variability (i.e., CV), and side asymmetry were computed. Table 3 provides an overview of the agreement of these mEar-derived gait metrics with the gold standard. Mean and variability of stride length yielded good to moderate agreement, while all other spatial gait cycle characteristics, particularly all parameters related to stride width, did not yield sufficient agreement with the gold standard.

### 3.5. Speed Dependence of Gait Segmentation

The relative errors (RSME_REL_) of the mEar-derived mean spatiotemporal gait parameters were further separately analyzed for each subject and the different walking speeds (slow, preferred, and fast) to investigate whether the performance of the gait identification algorithm depends on walking speed (Figure 3). Agreement of the mEar-derived gait measures was almost uniformly comparable across the three gait speeds, except for swing time, which showed a slightly larger deviation from the ground truth at slow walking speeds (F(2, 20) = 3.533, *p* = 0.042).

## 4. Discussion

This study aimed to explore mobile spatiotemporal gait characterization using a single ear-worn motion sensor. We developed and trained a deep-learning-based algorithm (mEar) based on gait measurements across a wide range of speeds, employing a large healthy cohort. mEar demonstrates high accuracy and good to excellent validity in characterizing a broad range of not only temporal but also spatial aspects of walking. This characterization remains largely consistent across various slow to fast walking speeds. mEar’s performance is comparable to leading algorithms that utilize multiple motion sensors on the lower extremities, showing only marginal differences in accuracy [13,14,24,25]. Compared to other body locations, the ear offers practical advantages for mobile health assessment due to its natural suitability for integrating wearable sensors with existing assistive devices such as eyeglass frames or hearing aids. Moreover, the ear’s anatomical location facilitates simultaneous monitoring of vital signs like pulse rate, blood pressure, oxygen saturation, and body temperature, complementing gait analysis [16,20]. This integration facilitates comprehensive health monitoring in daily life and has the potential to enhance telemedicine applications, highlighting the importance of gait as a general indicator of overall health.

Various methods have been proposed for segmenting gait patterns and characterizing spatiotemporal gait features using one or multiple body-fixed motion sensors in previous studies. These methods vary in accuracy and practicality, particularly in their ability to generalize across diverse measurement conditions. Signal processing techniques such as peak detection, template matching, and feature identification are commonly employed in these approaches [17,24,26]. However, they often rely on precise sensor positioning and orientation knowledge, necessitating careful calibration that can impede usability. Recently, deep-learning approaches have emerged as a promising alternative for mobile gait detection, offering enhanced robustness in segmenting gait patterns across different (noisy) measurement conditions compared to traditional methods [13,14,22,27]. In line with this, our deep-learning-based algorithm mEar demonstrates versatility by being indifferent to the specific ear side of sensor placement and operates effectively without initial calibration, accommodating variations in sensor orientation due to different ear anatomies. This renders our algorithm especially user-friendly and lowers the barriers for future clinical applications.

In terms of accuracy, methodologies employing foot-mounted motion sensors and deep-learning-based detection algorithms currently offer the most precise spatiotemporal gait characterization in everyday scenarios [13,22]. mEar achieves comparable accuracy in temporal characterization (step detection, stride time, and gait phases), demonstrating nearly flawless and highly precise temporal step detection across a broad spectrum of walking speeds. Despite the anatomical distance between the head and feet, our algorithm also provides accurate spatial characterization of stride length, albeit less precise than results obtained with foot-mounted multi-sensor systems [28]. Notably, mEar encounters challenges in accurately characterizing lateral gait, specifically stride width, similar to limitations observed in prior studies using multiple foot-mounted motion sensors [25]. Therefore, accurately characterizing spatial gait characteristics in the frontal plane remains challenging when relying solely on body-worn motion sensors.

It has been established that the quality of and impairments in walking are to be evaluated across five gait domains [29]: pace (e.g., walking speed and stride length), rhythm (e.g., swing and double support phases), variability (e.g., spatiotemporal stride variability), asymmetry (e.g., spatiotemporal stride asymmetry), and postural control (e.g., stride width). We have shown that mEar allows for the characterization of a range of spatiotemporal gait parameters with moderate to excellent concurrent validity (e.g., stride time, swing time, stride length, stride time variability, and stride time asymmetry), reliably covering all these essential dimensions of gait assessment except for the domain of postural control. The demonstrated accuracy of the spatiotemporal gait readouts from mEar may also allow the monitoring of gait changes in patients with clinically meaningful precision. For instance, the minimal clinically important difference (MCID) for gait speed across various clinical populations has been estimated to fall between 10 and 20 cm/s [30]. Assuming an average stride cycle lasts about 1 s, this would translate to a stride length MCID of approximately 10 cm, which is slightly above the observed RMSE_ABS_ for mEar-derived stride length, 9.7 cm. Beyond gait speed, changes in gait variability have been shown to provide important insights into fall risk and disease progression in conditions like cerebellar gait ataxia or Parkinson’s disease [31,32,33]. A MCID of 0.7% for temporal gait variability has been recently estimated for patients with Parkinson’s disease [34], which lies considerably above the precision of mEar-derived stride time variability with a RMSE_ABS_ of 0.3%. Finally, the MCID for gait asymmetry—a crucial metric for evaluating rehabilitation outcomes in stroke patients—has been recently estimated to be between 10 to 20%, which is well met by the precision of mEar-derived stride time asymmetry (RMSE_ABS_ of 0.3%) or stride length asymmetry (RMSE_ABS_ of 1.4%).

Mobile ear-based mobility and gait analysis can be seamlessly integrated into existing ear-mounted wearable technology for monitoring vital functions (Figure 4). The motion sensor used in this study is part of a wearable in-ear vital-sign monitor that allows continuous measurement of pulse rate, body temperature, and oxygen saturation. The integrated monitoring of activity (including gait) and vital signs holds significant potential for telemedicine applications in healthcare, as the data from both modalities can complement and enhance each other. Activity-aware vital-sign monitoring, on the one hand, enables the contextualization of patients’ vital functions based on their current physical activity (e.g., resting heart rate during inactivity, increased heart rate during physical exertion) [16,35,36,37]. This approach may help establish individual baselines for vital functions and improve the sensitivity of detecting anomalies that could indicate health issues. Conversely, vital-status-aware monitoring of gait allows the assessment of walking in the context of associated vital functions, providing crucial insights into the energy efficiency and economy of walking [38]. Continuous assessment of gait efficiency is particularly valuable in rehabilitation settings, as it can be used to adjust personalized recovery plans, thereby optimizing rehabilitation outcomes [39,40].

The here-proposed gait identification algorithm, mEar, possesses inherent limitations that need to be addressed in follow-up studies. Unlike stationary optical gait analysis methods, gait segmentation algorithms based on body-fixed motion sensors are not agnostic regarding the underlying motion sequence, and their detection accuracy can critically depend on the quality of the measured gait sequence or fail in cases of significantly altered gait patterns as observed in certain clinical populations. Therefore, the accuracies reported in this study, based on a healthy cohort and laboratory assessment settings with straight walking paths, may not directly translate to clinical populations. Moreover, the algorithm’s performance in unrestricted real-world environments remains uncertain due to its training conditions in a controlled laboratory environment. However, prior research from similar studies indicates that deep-learning algorithms for motion-sensor-based gait identification can be successfully adapted to clinical settings and real-world walking scenarios.

## 5. Conclusions

In this study, we introduced mEar, an algorithm for mobile gait analysis based on an ear-worn motion sensor. mEar allows for the precise determination of essential gait characteristics, including not only the average spatiotemporal gait pattern but also stride-to-stride variability and gait asymmetries. Thanks to its deep-learning-based architecture, mEar functions independently of which ear the sensor is worn on and requires no initial calibration regarding natural variations in sensor orientation. Combined with an in-ear vital-sign monitor, mEar enables parallel and complementary monitoring of activity and vital functions, presenting promising applications in telemedicine and rehabilitation. Further studies are, however, necessary to validate the algorithm’s effectiveness in clinical cohorts and real-life conditions.

## Figures and Tables

**Figure 1 sensors-24-06442-f001:**
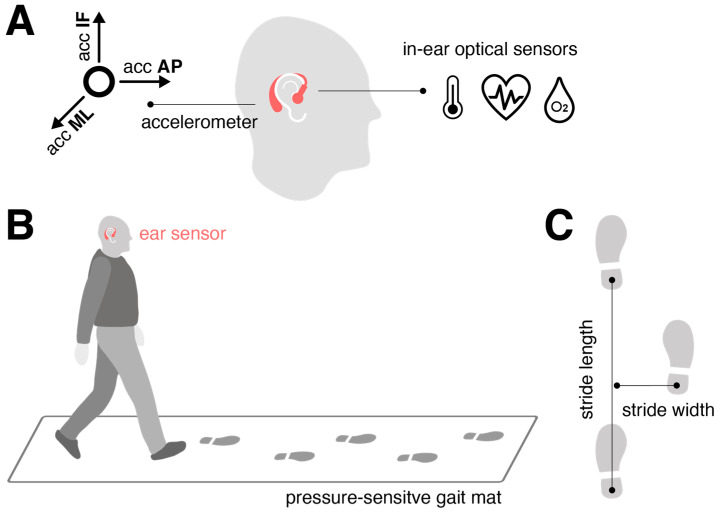
Experimental setup and analysis approach. (**A**) The motion sensor used in this study is integrated into an in-ear vital-sign monitor: the triaxial accelerometer is embedded in the processing unit behind the auricle, while the optical sensors for vital-sign monitoring are located on the sensor earplug that goes into the outer ear canal. (**B**) The participants walked across a 6.7 m pressure-sensitive gait mat synchronized with the ear-worn motion sensor at slow, preferred, and fast walking speeds. (**C**) Definition of spatial gait characteristics: Stride length is defined as the distance between two successive heel contacts of the same foot; stride width is defined as the perpendicular distance of one heel contact to the line connecting two successive heel contacts of the opposite foot. Abbreviations: acc: acceleration; ML: medio-lateral axis; AP: anterior–posterior axis; IF: inferior–superior axis.

**Figure 2 sensors-24-06442-f002:**
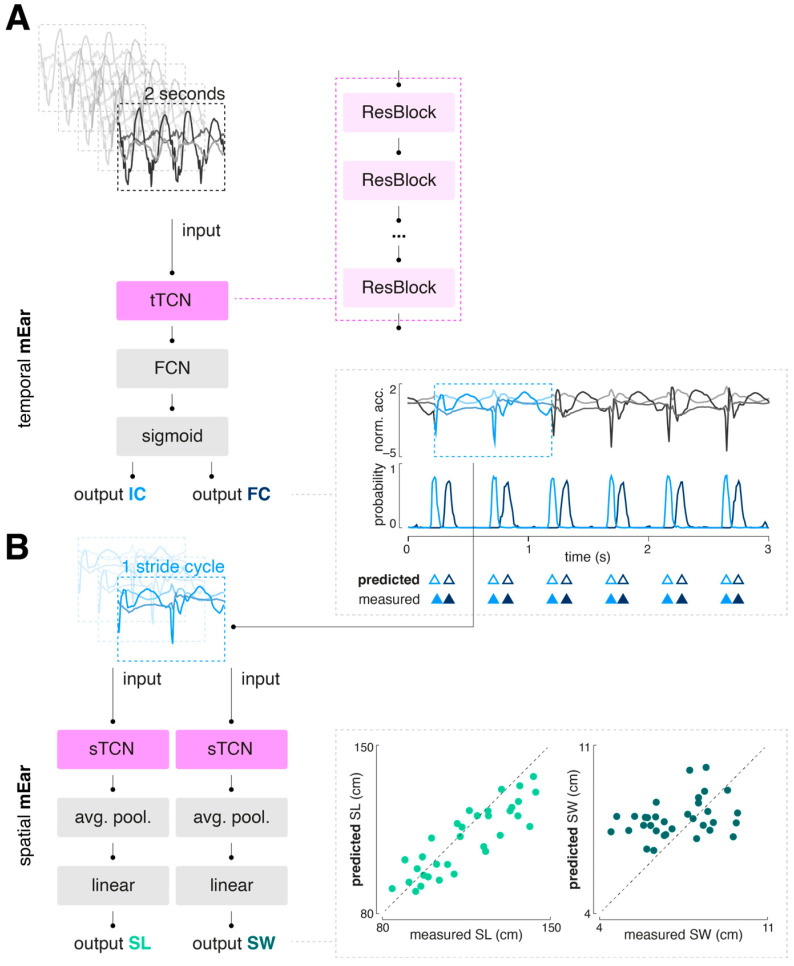
Overall model architecture of mEar. (**A**) The temporal mEar model processes raw triaxial accelerometer data (3 × 200 samples) to identify temporal gait events, i.e., ICs and FCs. It features a TCN architecture (tTCN) with residual blocks that have exponentially increasing dilation factors. Each residual block includes dilated convolutional layers, batch normalization, ReLU activation, and dropout layers. The TCN output is fed into an FCN with a sigmoid activation layer. (**B**) The identified IC events are used to extract accelerometer values for a full gait cycle, which are input for spatial-stride-characteristic regression using two spatial mEar networks. Both spatial models consist of a TCN architecture (sTCN), whose outputs are fed into an average pooling layer followed by a linear layer. Abbreviations: TCN: temporal convolutional network; FCN: fully connected layer; ReLU: rectified linear unit;; avg. pool.: average pooling; IC: initial contact; FC: final contact; SL: stride length; SW: stride width.

**Figure 3 sensors-24-06442-f003:**
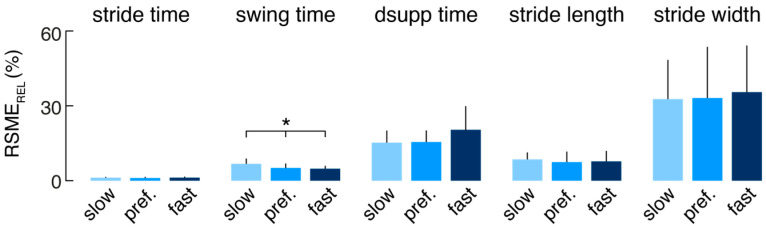
Agreement of estimated mean spatiotemporal gait cycle parameters at different walking speeds (* indicates a significant difference). Abbreviations: RMSE_REL_: relative root-mean-square error; dsupp time: double support time; pref.: preferred.

**Figure 4 sensors-24-06442-f004:**
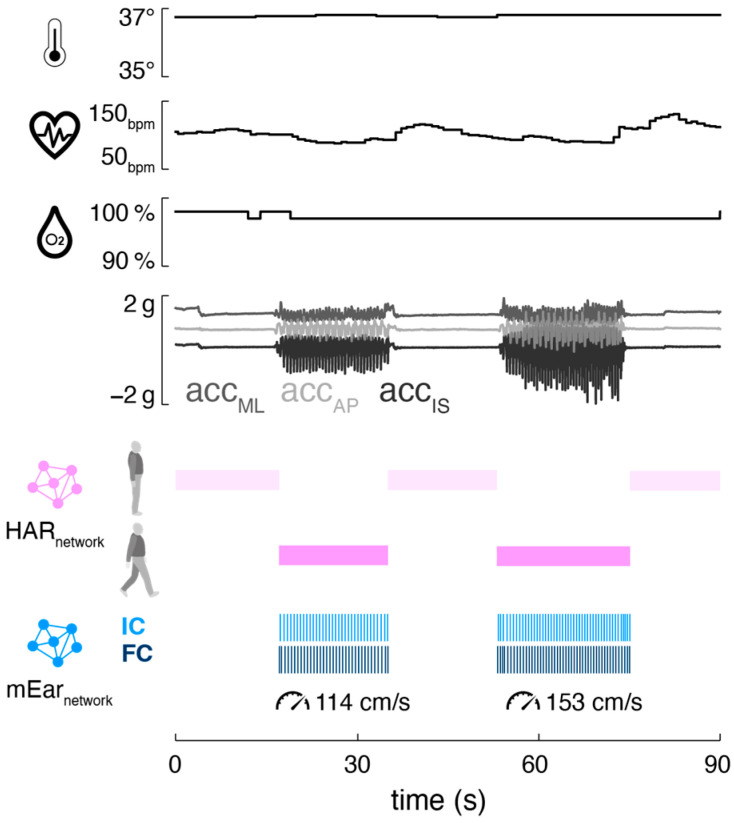
Integrated activity, gait, and vital-sign monitoring from a single ear-worn wearable. Exemplary activity [16] and gait classification output (temporal and spatial mEar) alongside vital-sign parameters (body temperature, pulse rate, and oxygen saturation) derived from a person standing and walking outdoors (first walking bout: stride time 1.1 s, stride length 122.5 cm; second walking bout: 0.9 s, stride length 142 cm). Abbreviations: bpm: beats per minute; acc: acceleration; ML: medio-lateral axis; AP: anterior–posterior axis; IF: inferior–superior axis; HAR: human activity recognition; IC: initial contact; FC: final contact.

**Table 1 sensors-24-06442-t001:** Detection performance and time agreement of predicted initial and final foot contacts of mEar compared to the gold standard (gait mat).

Parameter	TPs	FNs	FPs	Recall	Precision	F1	Time Error
**Initial contact**	3642	8	44	0.997	0.993	0.994	0.003 s
**Final contact**	3097	10	569	0.996	0.849	0.914	−0.002 s

Abbreviations: TPs: true positives; FNs: false negatives; FPs: false positives; F1: F1 score.

**Table 2 sensors-24-06442-t002:** Accuracy statistics of estimated temporal gait cycle parameters compared to the gold standard (gait mat).

Param.	Metric	mEar	Gait Mat	RMSE_ABS_	RMSE_REL_	R	ICC(3,1)
**Stride** **time**	Mean	1.2 ± 0.2 s	1.2 ± 0.2 s	0.0 s	0.3%	**0.999**	0.999
	CV	2.9 ± 2.4%	2.9 ± 2.5%	0.3%	10.0%	**0.993**	0.993
	Asym.	0.4 ± 0.3%	0.3 ± 0.3%	0.3%	0.9%	**0.635**	0.629
**Swing** **time**	Mean	0.4 ± 0.1 s	0.4 ± 0.1 s	0.0 s	3.8%	**0.968**	0.960
	CV	4.7 ± 1.7%	3.9 ± 1.5%	1.3%	34.5%	**0.814**	0.801
	Asym.	1.8 ± 1.5 %	0.9 ± 1.0%	1.7%	1.9%	**0.373**	0.347
**Dsupp** **time**	Mean	0.1 ± 0.0 s	0.2 ± 0.1 s	0.0 s	11.5%	**0.953**	0.944
	CV	12.6 ± 5.8%	10.2 ± 4.6%	5.5%	53.5%	**0.575**	0.559
	Asym.	3.1 ± 4.5%	2.2 ± 3.1%	5.8%	2.6%	–	–

Significant outcomes are marked in bold. Abbreviations: Dsupp time: double support time; CV: coefficient of variation; Asym.: asymmetry; R: Pearson’s correlation coefficient (significant results marked in bold); RMSE: absolute and relative root-mean-square error; ICC: intraclass correlation coefficient.

**Table 3 sensors-24-06442-t003:** Accuracy statistics of estimated spatial gait cycle parameters compared to the gold standard (gait mat).

Param.	Metric	mEar	Gait Mat	RMSE_ABS_	RMSE_REL_	R	ICC(3,1)
**Stride** **length**	Mean	111.0 ± 13.8 cm	115.1 ± 17.5 cm	9.7 cm	8.5%	**0.867**	0.843
	CV	4.4 ± 1.9%	2.7 ± 1.6%	2.2%	82.2%	**0.622**	0.617
	Asym.	1.1 ± 1.1%	0.3 ± 0.2%	1.4%	4.9%	0.081	0.035
**Stride** **width**	Mean	8.0 ± 0.8 cm	7.2 ± 1.4 cm	1.6 cm	21.7%	0.321	0.270
	CV	8.8 ± 5.6%	22.1 ± 8.8%	16.9%	75.7%	0.096	0.087
	Asym.	2.5 ± 2.4%	2.1 ± 2.3%	3.2%	1.5%	0.112	0.112

Significant outcomes are marked in bold. Abbreviations: CV: coefficient of variation; Asym.: asymmetry; R: Pearson’s correlation coefficient (significant results marked in bold); RMSE_ABS, REL_: absolute and relative root-mean-square error; ICC: intraclass correlation coefficient.

## Data Availability

Sample datasets, temporal and spatial mEar gait prediction models, and the corresponding scripts used in this study are publicly available at https://github.com/DSGZ-MotionLab/mEar, accessed on 23 September 2024. The complete data from this study can be obtained upon reasonable request from M.W. The participants did not consent to the publication of their sensor data in open repositories, in accordance with European data protection laws.

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
