# Peer review of "Mobile Spatiotemporal Gait Segmentation Using an Ear-Worn Motion Sensor and Deep Learning"

_sensors, 2024, doi:10.3390/s24196442_

Round 1

Reviewer 1 Report

Comments and Suggestions for Authors

Dear authors, dear Julian,

I would like to thank you for this superb research. I have read the manuscript with great pleasure and think the conducted study fits well within trends to move to mobile gait analysis. I agree that an ear-worn sensor has received little attention, although there is great potential given the ubiquitous use of glasses and hearing aids among the older population. The results look promising and you have acknowledged that your model may need additional validation or training for clinical populations.

To be honest, I don’t really see anything that needs to be adjusted per se, but I do have some questions, out of pure curiosity, that you may address or discuss in the manuscript.

Q1. You have used a single ear-worn accelerometer. If the participants wore the device on their right ear, did you then observe that the right gait events and derived spatiotemporal gait parameters were more accurate than the left event and parameters?

Q2. Why did you use train two separate models for the spatial parameters rather than one model that predicts both stride length and width? Did you observe that the less accurate stride width predictions corresponded also to the less accurate stride length predictions?

Q3. Will the models and scripts be available, for example on GitHub?

Author Response

Dear authors, dear Julian,

I would like to thank you for this superb research. I have read the manuscript with great pleasure and think the conducted study fits well within trends to move to mobile gait analysis. I agree that an ear-worn sensor has received little attention, although there is great potential given the ubiquitous use of glasses and hearing aids among the older population. The results look promising and you have acknowledged that your model may need additional validation or training for clinical populations.

To be honest, I don’t really see anything that needs to be adjusted per se, but I do have some questions, out of pure curiosity, that you may address or discuss in the manuscript.

Response: Thank you very much for the review and the favorable feedback on our manuscript.

Comment: You have used a single ear-worn accelerometer. If the participants wore the device on their right ear, did you then observe that the right gait events and derived spatiotemporal gait parameters were more accurate than the left event and parameters?

Response: Thank you for the question. We do not observe any differences or side preference in the quality of gait prediction, which would also be expected since the sensors on the left and right ears both measure the same rigid body kinematics (the head), and therefore provide more or less identical information.

Comment: Why did you use train two separate models for the spatial parameters rather than one model that predicts both stride length and width? Did you observe that the less accurate stride width predictions corresponded also to the less accurate stride length predictions?

Response: Thank you very much for the question. Indeed, we initially also tried to predict the spatial parameters with a single model. However, the results were significantly less accurate compared to the approach ultimately used with two separate models.

Comment: Will the models and scripts be available, for example on GitHub?

Response: Yes, we will publish the models and the corresponding scripts, including a sample dataset, on GitHub (https://github.com/DSGZ-MotionLab/mEar). The link to the repository will also be provided in the final manuscript.

Reviewer 2 Report

Comments and Suggestions for Authors

The study is new and useful for utilizing an ear-worn motion sensor for spatiotemporal segmentation of gait pattern. Different from the existing method, the proposed method can focus on a viable site for monitoring gait and manifest the potential integration to in-ear vital sign monitoring to some extent. I believe that the integration is required to offer a feasible approach to understanding health monitoring and telemedical applications. I thus recommend to accept the study to publication if the following major revision is yielded.

1) The authors used a modified temporal convolutional network (TCN) architecture to construct their classifier from sampling data to gait patterns, and trained a total of three neural networks. But some key information fails, such as the loss function, clear network structures and so on. I thus suggest that the authors use a table to illustrate these key factors such as parameters, layers etc. In addition, the authors state they have revised the TCN, and future summarize the steps for modification.

2) Theirs research samples come from 53 adults, but it is unclear whether their results and methods are suitable for other populations such as children, and whether the gender and other characteristics of adults are also specified, please?. Also, can the current zero sample learning and transfer learning be used to further enrich the model?

Comments on the Quality of English Language

Some revisions are required to make.

Author Response

The study is new and useful for utilizing an ear-worn motion sensor for spatiotemporal segmentation of gait pattern. Different from the existing method, the proposed method can focus on a viable site for monitoring gait and manifest the potential integration to in-ear vital sign monitoring to some extent. I believe that the integration is required to offer a feasible approach to understanding health monitoring and telemedical applications. I thus recommend to accept the study to publication if the following major revision is yielded.

Response: Thank you very much for reviewing and for the positive feedback on our manuscript.

Comment: The authors used a modified temporal convolutional network (TCN) architecture to construct their classifier from sampling data to gait patterns, and trained a total of three neural networks. But some key information fails, such as the loss function, clear network structures and so on. I thus suggest that the authors use a table to illustrate these key factors such as parameters, layers etc. In addition, the authors state they have revised the TCN, and future summarize the steps for modification.

Response: Thank you for the comment. Details regarding the model training, such as the loss function used, are already provided in the methods section of the manuscript (lines 172-189). However, we agree that a comprehensive tabular overview of the model architectures is useful, and we have therefore included this information in the form of supplementary tables in the revised manuscript.

Comment: Theirs research samples come from 53 adults, but it is unclear whether their results and methods are suitable for other populations such as children, and whether the gender and other characteristics of adults are also specified, please? Also, can the current zero sample learning and transfer learning be used to further enrich the model?

Response: Thank you for the comment. We fully agree and have addressed this limitation in the discussion (lines 374-386), noting that the current model’s gait prediction results cannot be easily generalized to other cohorts, such as children, the elderly, or clinical populations with neuro-geriatric gait disorders. As detailed in the methods section, our cohort consists of 53 healthy individuals (ages 20–59) with a balanced gender ratio (53% women). At this stage, we are unable to confirm if the trained algorithm can accurately identify pediatric gait. However, we are currently using the sensor system to record clinical (Parkinsonian and ataxic gait patterns) and geriatric gait data. Once recruitment is completed, we will evaluate the model’s performance on these gait patterns and determine if additional steps, such as model generalization through transfer learning, are necessary. As noted in the discussion, other studies suggest that gait recognition using body-fixed sensors generally transfers well from healthy cohorts to geriatric or clinical populations (e.g. Romijnders et al. Front Neurol 2023; Seifer et al. IEEE J. Biomed. Health Inform 2024).

Round 2

Reviewer 2 Report

Comments and Suggestions for Authors

I compared this new version with the original version, and my concerns were almost not fully addressed and resolved, such as the  organization, representativeness, and sufficiency of the training samples used, the constructed network structure and its basic parameters settings. These factors have a significant impact on the application and generalization ability of the results, but the author did not provide a convincing result at all. I think it still is required to be further revised to reach the average level of the journal and may be useful to readers or users. Overall, I do not support the publication of this mc in its current form.

Comments on the Quality of English Language

ok

Author Response

Comment: I compared this new version with the original version, and my concerns were almost not fully addressed and resolved, such as the organization, representativeness, and sufficiency of the training samples used, the constructed network structure and its basic parameters settings. These factors have a significant impact on the application and generalization ability of the results, but the author did not provide a convincing result at all. I think it still is required to be further revised to reach the average level of the journal and may be useful to readers or users. Overall, I do not support the publication of this mc in its current form.

Response:
Thank you for your continued feedback on our manuscript. We appreciate the time and effort you’ve taken to review the revised version. In response to your comments, we would like to provide clarification on the revisions we have made, both following the first round of review and with this latest version.

Already in response to the first round of review, we aimed to address all missing information regarding the network structure by providing detailed supplementary tables (Tables S1 and S2). Furthermore, we made the complete models and all scripts used for analysis available in a public GitHub repository, referenced in the manuscript, to ensure transparency and reproducibility.

In this latest revision, we have taken additional steps to enhance the clarity of our work. We have now included a new figure (Figure S1) that outlines the exact structure of each model, displaying the corresponding graph for each. We outlined the specific options and results for model optimization via hyperparameter grid search (Table S3). Additionally, beyond the detailed dataset information already provided in the first paragraph of the Results section, we now explicitly report the exact training and test sample sizes for each model.

Given these comprehensive revisions, we currently feel that all necessary information to ensure transparency and reproducibility has been included. We believe the current manuscript, along with the supplementary materials and publicly available resources, provides all necessary details for replicating and understanding the methodology and results presented. We hope this clarifies our approach and the steps we’ve taken to meet the journal’s standards and ensure transparency for the readers.

Round 3

Reviewer 2 Report

Comments and Suggestions for Authors

I have read the author's response and further improvements, and have evaluated the overall value of the modified research results in the mc. I nearly believe that the results provide clear explanations for the extracted human feature points and provide a comprehensive analysis of the experimental results, which have reference value for user of potiental peers. Their findings are beneficial for both healthcare and training guidance.
Although I am not entirely satisfied with the overall article, considering the timeliness of the author's research results, I agree to accept its publication. I hope the author can address and clarify all the issues I previously raised in future work.

Comments on the Quality of English Language

OK